# Wide-Field Calcium Imaging of Neuronal Network Dynamics In Vivo

**DOI:** 10.3390/biology11111601

**Published:** 2022-11-01

**Authors:** Angela K. Nietz, Laurentiu S. Popa, Martha L. Streng, Russell E. Carter, Suhasa B. Kodandaramaiah, Timothy J. Ebner

**Affiliations:** 1Department of Neuroscience, University of Minnesota, Minneapolis, MN 55455, USA; 2Department of Mechanical Engineering, University of Minnesota, Minneapolis, MN 55455, USA

**Keywords:** mesoscopic imaging, calcium imaging, neuronal dynamics, functional connectivity, spatial independent component analysis

## Abstract

**Simple Summary:**

We review advances in the properties of calcium sensors, microscopy, and data analysis that have brought about the ability to image large contiguous regions of the mouse brain, often termed wide-field or mesoscale imaging. We summarize representative wide-field imaging studies spanning several neuroscience subfields, providing an overview of insights gained into brain function. Finally, we present new developments in wide-field imaging that allow for more comprehensive investigation of activity across the brain.

**Abstract:**

A central tenet of neuroscience is that sensory, motor, and cognitive behaviors are generated by the communications and interactions among neurons, distributed within and across anatomically and functionally distinct brain regions. Therefore, to decipher how the brain plans, learns, and executes behaviors requires characterizing neuronal activity at multiple spatial and temporal scales. This includes simultaneously recording neuronal dynamics at the mesoscale level to understand the interactions among brain regions during different behavioral and brain states. Wide-field Ca^2+^ imaging, which uses single photon excitation and improved genetically encoded Ca^2+^ indicators, allows for simultaneous recordings of large brain areas and is proving to be a powerful tool to study neuronal activity at the mesoscopic scale in behaving animals. This review details the techniques used for wide-field Ca^2+^ imaging and the various approaches employed for the analyses of the rich neuronal-behavioral data sets obtained. Also discussed is how wide-field Ca^2+^ imaging is providing novel insights into both normal and altered neural processing in disease. Finally, we examine the limitations of the approach and new developments in wide-field Ca^2+^ imaging that are bringing new capabilities to this important technique for investigating large-scale neuronal dynamics.

## 1. Introduction

Behavior involves processing and integrating information within and across brain regions. However, the mechanisms by which neuronal activity is coordinated to produce a unitary behavioral output are currently not well understood. A prominent view is that sensory, motor, and cognitive behaviors are the emergent properties arising from the interactions among neurons [1,2,3,4]. It is also widely acknowledged that deciphering how brain structures, such as the cerebral cortex, basal ganglia and cerebellum, plan, generate, and acquire behaviors requires descriptions of the neuronal activity at multiple scales [5,6,7,8]. Spatially, the scales range from activity across different cortical regions, to neuronal, and even subcellular levels.

By taking advantage of the modular organization and segregation of brain regions into anatomically and functionally distinct regions, neuroscientists have made enormous progress in understanding how computations in different areas are involved in behavior. However, the operations of the brain cannot be easily understood by only studying its various regions in isolation. For example, in the cerebral cortex, which will be the focus of this review, even simple sensory and motor tasks involve processing of information in multiple cortical areas. Deflection of a single whisker results in activation distributed across the sensorimotor cortices [9]. Locomotion activates much of the cerebral cortex, including extensive modulation of neuronal responsiveness in the primary visual cortex, even in the dark [10,11,12]. Arousal and attention exert markedly different effects across the cerebral cortex, both spatially and temporally [13,14,15]. Information flow across the cortex is dependent on the internal brain state, behavioral context, and past experiences [12,14,16,17]. Furthermore, learning a new task produces widespread changes in cortical dynamics [18].

Therefore, understanding cerebral cortical function in behavior and in different brain states requires a mesoscopic level description of the cortical regions engaged and their interactions. In the nervous system, mesoscopic refers to structure and function between the microscopic and macroscopic levels, that is between the levels of single neurons and entire brain regions, respectively. In physical systems, new properties emerge at the macroscopic level that are not observable at smaller scales. At the macroscopic level, temperature, viscosity, and density are the collective properties of the statistical motions of the atoms and molecules of gases and fluids, not of the individual particles. Magnetism and conductivity do not exist at the atomic level, instead emerging from the interactions among individual atoms (technically these are examples of weak emergence). Similarly, in the nervous system, it is the interactions among populations of neurons that underlies the neural representations of perception, behavior, and brain states. Therefore, understanding neuronal dynamics at the mesoscale are of fundamental importance.

Originally, wide-field imaging relied on a blood flow/oxygenation-related intrinsic optical signal, voltage sensitive dyes, or flavoprotein autofluorescence [19,20,21]. These methods were limited by low signal-to-noise ratio and/or slow readouts as well as limited specificity [22,23]. The development of wide-field Ca^2+^ imaging with genetically encoded Ca^2+^ indicators (GECIs) overcame these limitations and has allowed simultaneous imaging of neuronal activity over large regions at relatively high spatial and temporal resolution. Ca^2+^ imaging has been extensively used in in vitro systems, particularly neuronal cultures, to study the development and functional organization of de novo network formation and test the properties of Ca^2+^ sensors [24,25], and in vivo to study intrinsic network dynamics. More recently, wide-field Ca^2+^ imaging is being explored in different species, including rats [26,27], marmosets [28,29,30], and macaques [31,32], with an emphasis on single cell resolution. This review focuses on awake, in vivo mesoscopic imaging in mice during behaviors as a tool to understand the neuronal dynamics of sensation, action, and cognition.

## 2. Visualizing Neural Activity with Ca^2+^ Sensors

Measurement of intracellular Ca^2+^ concentration has long been a gold standard for optical monitoring of neural activity. Ca^2+^ concentration provides a fast readout of changing neural activity, although indirect, as it is involved with numerous intracellular signaling cascades, intracellular Ca^2+^ storage and release, and synaptic transmission [33,34,35,36]. Some of the earliest Ca^2+^ sensors were dyes, such as BAPTA, fura-2, and fluo-4 [36]. While these sensors have fast kinetics with bright fluorescence, the dyes are primarily used for acute imaging studies as they are not retained inside cells for long time periods.

Based on the Ca^2+^ binding protein calmodulin, GECIs have been developed that allow for chronic imaging of neural activity in vivo [35,37,38,39,40]. The most common of these is GCaMP, which is a fusion protein composed of calmodulin as the sensor to monitor changes in Ca^2+^ concentration, the M13 sequence of the myosin light-chain kinase, and a circularly permuted GFP molecule that acts as the fluorescent reporter whose intensity is the readout of changing Ca^2+^ concentration [34,38,39,40]. GECIs have many advantages over chemical Ca^2+^ sensors, as the genes for the various proteins can be inserted into the genome or transfected into cells for stable, uniform, long-term expression and chronic monitoring of neural activity in vivo [41,42]. Additionally, dual expression systems in combination with intersectional genetics and/or promoter specific viruses can be used to express sensors in specific populations or subpopulations of cells [43,44,45]. Utilization of an ever-expanding number of Cre-driver lines has revealed a litany of subtle response properties in visual cortical neurons [46]. Conversely, strategies for driving sparse expression of Ca^2+^ indicators allow for single cell resolution at the mesoscale level [47,48].

Targeted mutations have improved the Ca^2+^ affinity, kinetics, and fluorescence brightness of GCaMPs with the widely-used GCaMP6f capable of detecting single action potentials [33,38], with further improvements provided by the newer GCaMP7 and 8 families of GECIs [37,49]. Coupling with other fluorophores, such as mRuby and mApple, have yielded red-shifted RCaMPs with less phototoxicity, increased imaging depth in tissues, and which can be used in conjunction with optogenetics and other GFP-based reporter proteins [50]. To track specific cell populations with high resolution, photoactivatable versions of both red and green Ca^2+^ sensors are now available, in which colored light induces a state switch from low/no fluorescence to high fluorescence [51,52]. Another important development has been in viral vectors that cross the blood–brain barrier for widespread expression of genetically encoded sensors in the CNS via systemic delivery routes [53,54]. These advances in biosensor properties, expression, and delivery methods allow for improved and targeted imaging to investigate neural circuitry and activity.

## 3. Monitoring Wide-Field Ca^2+^ Imaging Dynamics In Vivo

The improvements in GECIs have made mesoscopic imaging of neural circuits in awake behaving animals possible. Imaging is typically performed with the animal head-fixed after the implantation of a cranial window, in either a closed- or open-skull format [55]. Closed-skull imaging often involves thinning of the overlaying bone and/or the use of optical clearing solutions to achieve optical clarity and refractive index matching [56,57]. Partial transparency can be achieved through full-thickness skull with the use of glass coverslips [58,59]. Most open-skull window techniques involve either a large or small craniotomy and replacing the bone flap with either glass or biocompatible polymers [55,60,61,62]. Thinned or intact-skull preparations offer the advantage of visualizing skull landmarks for easier comparisons between subjects and between recording sessions, and are minimally invasive. However, imaging through the skull decreases spatial resolution and imaging depth. Open-skull preparations with windows, while invasive, offer high resolution imaging and better tissue penetration.

The technical aspects of wide-field Ca^2+^ imaging are relatively straight-forward, as they are based on single photon fluorescence imaging. Whether using a custom built or commercial system, most use low magnification optics and lenses. Excitation light is typically provided by LEDs and emission fluorescence is captured with a scientific CMOS camera (e.g., Orca from Hamamatsu, Hamamatsu-shi, Japan or Andor Zyla from Oxford Instruments, Belfast, UK). Newer scientific CMOS cameras have excellent sensitivity, high quantum efficiency, low noise, and fast frame rates, even with large sensor formats. Due to the kinetics of GCaMP6f, typical acquisition frame rates have been between 20–40 Hz. Wide-field Ca^2+^ imaging often requires animal restraint due to the size of the microscope and objective as well as need to stabilize the field-of-view. For this reason, a head fixation post, such as a titanium head plate is implanted with the cranial window [60,63]. Imaging can then be performed at rest using a restraint tube or in behavioral paradigm setups modified for head-fixation, for example a spherical or disk treadmill to accommodate locomotion (Figure 1A,D) [61,63,64].

Imaging of GCaMP Ca^2+^ fluorescence with blue light (~470 nm) yields fluorescence in the green spectrum (~510 nm). Blood flow increases with neuronal activation, and oxygenated blood absorbs light with peak absorption at ~530 nm, resulting in a darkening that decreases the GCaMP fluorescence [8,65]. Therefore, the effects of blood flow and other Ca^2+^-independent fluorescence changes, such as flavoprotein autofluorescence, should ideally be removed. Multiple methods have been employed to correct for changes in blood flow, with interleaved dual wavelength imaging becoming the preferred approach (Figure 1B,C) [8,22,61,66,67]. For epifluorescence dual wavelength imaging, a wavelength at the isosbestic point for GCaMP that is Ca^2+^-independent (~405 nm) is interleaved with the excitation wavelength, correcting the Ca^2+^-dependent GCaMP signal by scaling and subtracting the blood flow signal. Dual wavelength reflection imaging based at the isosbestic point for hemoglobin has also been used [65,67,68]. Other methods for hemodynamic correction include filtering of the GCaMP signal [69], using independent component analysis (ICA) [18], and masking of the vasculature [70].

## 4. Limitations of Wide-Field Imaging

As with any technique, mesoscopic imaging has limitations. Based on single photon (1P) epifluorescence imaging, the mesoscale approach is restricted to recording activity near and below the surface of brain. For example, in the cerebral cortex, the Ca^2+^ fluorescence signals are primarily from neuronal activity in layers II/III [65,71]. The depth of recording for 1P imaging in dense tissues can be improved, to some degree, with red-shifted sensors [50,72], as longer wavelengths penetrate tissue better and are prone to less scatter. In some small systems with unimpeded optical access, such as larval zebrafish, imaging the entire brain using light sheet microscopy can allow for volumetric functional wide-field single-cell Ca^2+^ imaging [73,74]. This is extremely challenging to perform in systems other than zebrafish or fixed tissue.

Due to light scatter produced by imaging through dense tissue and the low numerical aperture of mesoscopic lenses, mesoscopic imaging cannot provide high lateral or axial resolution as with other forms of microscopy [75]. Instead, the observed neural signal is a population summation of the activity within essentially a 3-dimensional voxel (not just 2 dimensional) that includes the dendrites, somata, and axons within that volume. While typical wide-field Ca^2+^ imaging does not report the activity of single cells, there are strategies, such as sparse expression of indicators and/or restriction to the soma, that can provide single cell resolution [76,77].

In addition, the temporal resolution of mesoscale imaging is limited by the kinetics of the GCaMP protein. While the sensitivity and speed of GECIs have been vastly improved in recent years with targeted mutations, their decay time for single action potentials is still greater than 100 ms and their ability resolve high frequency firing with concurrent increases in fluorescence amplitude is also limited by both saturation and kinetics [38,50,75]. The timescale of GCaMPs are considerably slower than those of action potentials, which are on the order of 2–5 ms [78]. This difference in timescale between action potentials and Ca^2+^ fluorescence, especially in cases where single cells can be isolated, has led to the development of algorithms to reconstruct underlying spike trains from recorded Ca^2+^ signals [79,80,81] and methods of simultaneous electrophysiological and Ca^2+^ recordings [38,40]. The accuracy of inferred spike trains from Ca^2+^ data and construction of neural networks is dependent upon the method of reconstruction used and the experimental noise level [82,83]. These confounds can influence network connectivity and spike inference results and need to be considered during interpretation of results.

As discussed above, the primary family of Ca^2+^ indicator used, GCaMP, can be contaminated by both hemodynamics [8,65,67] and other Ca^2+^-independent fluorescence changes such as flavoprotein autofluorescence [58,66] requiring a correction strategy such as dual wavelength imaging. Additional contamination can arise from brain motion, baseline signal fluctuations, rhythmic physiological processes, and photonic noise (i.e., variability in the emission/detection of photons) [79]. Noise sources are a significant confounding factor in spike estimation, but newer algorithms are being improved to accommodate additional noise sources [84]. While the limitations of wide-field Ca^2+^ are not to be underestimated, its utility in revealing signaling relationships between brain areas remains invaluable.

## 5. Analysis of Wide-Field Ca^2+^ Imaging Data

### 5.1. Functional Segmentation of the Imaging Field

One challenging aspect of wide-field Ca^2+^ imaging is how to analyze and interpret the complex data sets collected. Numerous approaches have been utilized and developed over the years to investigate neuronal dynamics based on Ca^2+^ signals, and how those Ca^2+^ signals change over time, relate to behavior, or interact with signals from other brain regions [18,22,85]. An important step in wide-field imaging analysis is defining the regions. One of the main challenges is the multidimensional nature of the optical recordings. Each pixel is a mixture of multisource signals from dendrites, axons, and somata from different neurons as well as different cortical layers [22,71], and identifying meaningful, separable functional regions or cells requires decomposing the signal into its component parts.

Segmentation approaches range from manual parcellation of the brain to automated algorithms that aid in denoising, demixing, and segmenting the Ca^2+^ data in an unbiased manner. Many wide-field Ca^2+^ imaging studies have analyzed the change in fluorescence (ΔF/F) from regions of interest (ROIs) placed over the imaging field [86,87,88]. Placing multiple ROIs allows the interactions of several brain regions to be studied. The ROIs can be drawn manually (Figure 1B,C) or can be placed in an unbiased approach, such as using the borders of the Allen Brain Atlas Common Cortical Framework (CCF) [48,58,89]. Spatial maps of activity can also be obtained using seed-based correlation methods with or without co-localized tract-tracing [90,91,92] or spike-triggered mapping [58,69]. Clustering algorithms have also been used in segmenting both fMRI and Ca^2+^ imaging data by identifying common features of signals across recorded pixels [93,94,95].

Another approach to functional segmentation is blind source separation (BSS). Unsupervised and data-driven, BSS methods operate by decomposing the signal into sources that are statistically independent. Separation methods, such as principal component analysis (PCA) and singular value decomposition (SVD), have been applied to wide-field Ca^2+^ imaging data [17,96,97]. However, PCA/SVD require that the vector outputs are orthogonal, which may not be physiologically relevant. Furthermore, PCA/SVD analysis can result in de-localization of a spatial component, where the component encompasses multiple brain regions. This can make it difficult to interpret the spatial and temporal components, both within and between animals, without additional thresholds or restrictions of the spatial component.

Some BSS methods do not require orthogonality, such as independent component analysis (ICA), which separates the data into maximally statistically independent components (ICs) [98,99,100]. Several ICA algorithms, like JADE and FastICA, are based on high-order statistics [101], while others are based on second-order statistics using signal coherence [102] or intra-source decorrelation [103]. For data expressed in both the spatial and temporal dimensions, such as wide-field Ca^2+^ imaging, ICA can be applied in the spatial domain to reveal independent spatial components (spatial ICs) and the associated activity time-courses (Figure 2A). Alternatively, ICA can be applied to the temporal domain to reveal independent temporal patterns and their associated locations [98,104,105].

Another non-orthogonal BSS method is non-negative matrix factorization (NMF), which restricts the spatial and/or temporal components to be non-negative [107]. Different NMF algorithms have been used for Ca^2+^ imaging [106,108,109]. However, due to the complex nature of the Ca^2+^ temporal signal, a common approach is to normalize the raw fluorescence by mean-subtracting the signal (ΔF/F), which yields negative values. As such, NMF does not handle mixed-sign source signals well and can produce delocalized spatial components like some BSS methods. To address these issues, the non-negative restriction can be relaxed and the spatial component restricted to a common anatomical atlas, such as the Allen Brain Atlas Common Cortical Framework (CCF) or any other reference atlas of choice, to aide in anatomical segmentation of the spatial components [106]. This version of NMF, localized semi-NMF (LocaNMF), increases the interpretability and reproducibility of results, and the use of a reference atlas facilitates comparisons within and between subjects (Figure 2B). As for all BSS methods, once defined, the spatial components can be used to investigate region-based neuronal Ca^2+^ activity in a hypothesis-driven manner.

### 5.2. Ca^2+^ Imaging-Based Functional Connectivity

Once segmented, the relationship between Ca^2+^ fluorescence in different functional regions can be used to generate functional connectivity (FC) maps. This is typically done by summarizing Ca^2+^ activity relationships in a connectivity matrix, in which each entry is a pair-wise measure of the correlation between regions. The matrix is populated with a measure of correlation, with the Pearson correlation coefficient commonly used [61,90,110,111]. From here, graph theory analysis can be used to quantify the FC properties between brain regions and how they change in different conditions [112]. In this method, brain regions are nodes in the graph with functional connections represented by lines or edges (see Figure 3). The resulting graph reveals the functional topology and architecture of the network. A multitude of network properties and statistical comparisons can be calculated, at both a local and global scale, using available toolboxes [113,114,115].

### 5.3. Analyzing the Relationship between Ca^2+^ and Behavior

Of special interest are analytical methods that relate the neural activity to behavioral parameters, including movement kinematics, task performance, and/or decision-making. The simplest analysis compares the activation of the cortex in a specific behavioral context to a baseline. Commonly employed methods are based on linear models, such as multiple linear regression or general linear models (GLM), that correlate the Ca^2+^ fluorescence time courses of pixels or ROIs against the parameters of interest. These approaches can quantify the concurrent influences of multiple behavioral parameters and the temporal relationship between behavior and neural activity [8,17,95,116]. Linear models can also be used to relate cortical Ca^2+^ activity to electrophysiological recordings [97].

## 6. Insights Gained from Wide-Field Ca^2+^ Imaging

Over the last 15 years, an increasing number of publications have used wide-field Ca^2+^ imaging, primarily in the cerebral cortex, addressing multiple questions concerning action, sensory perception, and executive function. Here, we review select studies to illustrate the use of, and intriguing results in motor control, learning, decision-making, and neurological disorders gained from, mesoscopic Ca^2+^ imaging in the cerebral cortex.

### 6.1. Elucidating Motor Control Using Wide-Field Ca^2+^ Imaging

A major advantage of wide-field Ca^2+^ imaging is that it has allowed the characterization of widespread changes in cortical activity and interactions during behaviors that were once thought to be mediated largely by a few key brain regions. In fact, a hallmark of wide-field imaging studies has been that behaviors engage the entire cerebral cortex [8,17,109]. One example is in reach to grasp (RtG) behavior, which has largely been examined in the context of motor areas of the neocortex [109,117]. A recent study examined the wide-field dynamics of excitatory neocortical neurons during a task in which mice performed a reach for a food pellet reward [109]. Despite this RtG task requiring a unilateral limb movement, widespread, bilateral increases in activity were observed during RtG, beginning just prior to reaching onset. Large alterations in FC were also observed, with an increase in correlations across the neocortex around the time of movement onset that subsequently decreased during the actual RtG movement. Together, these results demonstrate that even discrete motor tasks like unilateral RtG involve coordinated, bilateral alterations in cortical activity and functional interactions throughout the cerebral cortex.

As noted in the Introduction, locomotion results in widespread changes in cerebral cortical neuronal activity. For example, during locomotion neural firing increases in the somatosensory cortex [118,119,120], and in the primary visual cortex [121,122], while neuronal firing decreases in the primary auditory cortex [123]. These firing modulations are internally driven by the locomotor state, as they do not depend on the arousal level, and can occur without changes in sensory input [124]. These global changes in activity are thought to optimize the behavior by updating brain regions on the ongoing locomotion and tune circuits accordingly [122,123].

We investigated the changes in cortical activity during spontaneous locomotion using Ca^2+^ imaging through transparent polymer skulls in GCaMP6f mice [61]. Functional connectivity (FC) was based on the correlations between time series of the changes in Ca^2+^ fluorescence from 28 regions (nodes) obtained using spatial ICA. The changes in FC were determined through a series of six behavior periods, spanning from rest, rest to locomotion, continued locomotion, and locomotion to rest. Compared to rest, the cerebral cortex enters a new state with a distinct pattern of interregional FC (Figure 3A). The correlations and centrality of nodes in the primary motor and somatosensory cortices decrease, while the correlations and centrality of the retrosplenial cortex increase (Figure 3A,B). The locomotion state is preceded and followed by transition states characterized by dramatic increases in FC across the cortex (Figure 3A(i,v)) and large shifts in centrality (Figure 3B(i,v)). Importantly, the correlations, centrality, and the outward causality of nodes in the anterior premotor M2 region increase at the onset of locomotion. The changes in FC are independent of the changes in fluorescence that occur during locomotion. These results highlight the transient changes in FC in the cerebral cortex, from rest to locomotion and on return to rest, and suggest a key role for the anterior premotor regions in this self-initiated movement.

### 6.2. Cortical Dynamics during Learning, Executive Functions, and Decision-Making

A predominant theory is that learning is mediated by changes in emergent properties of neuronal networks in the brain. However, most previous learning studies have been limited to individual brain regions. In addition to monitoring widespread network dynamics during behavior, wide-field imaging allows for the characterization of how these mesoscale-level dynamics are transformed over time. During a motor learning task in which mice learn to press a lever to a specific threshold after an auditory cue [18], a coordinated sequence of cortical activation emerges, beginning around the time of movement onset. After repeated trials over days, the reaching kinematics stabilized, indicative of a learning process. In this study, motor learning was associated with a compression of the cortical activation sequence, characterized by a decreased latency to activation of each region and a decrease in the time for that activation to spread throughout cortical regions. Intriguingly, as learning progressed, premotor cortical area M2 emerged as a directing node in the reach network as illustrated by an increase in Granger Causality (Figure 4A). This suggests that the execution of skilled tasks involves widespread, sequential activation of cortical regions, and that refinement of this process via learning is mediated by the stabilization and compression of these sequences, potentially directed by premotor regions.

Researchers are taking advantage of the ability to examine neural activity and interactions over large regions to probe cortical activity during decision-making, planning, and higher processing, using behavioral tasks modified for use with head fixation [117]. Virtual reality tools make implementing decision-making and other behavioral paradigms significantly easier by allowing the animal’s range of motion to remain limited as the animal navigates a simulated environment [66]. A common observation is task-engagement involves global activity changes across the neocortex [8,126]. Both virtual and real visual discrimination (go/no-go), odor-discrimination, and evidence-based tasks reveal that Ca^2+^ fluorescence ramps up across the cortex [8,95,126]. Several groups have shown that cortical regions enter a desynchronized state in response to task involvement and low-frequency brain oscillations are suppressed during these times (Figure 4B,C) [66,95]. The desynchronization appears to be related to the probability of movement. These global signal increases are not due to specific task-related events, but rather to task-performance in general. 

Wide-field Ca^2+^ imaging has been combined with optogenetics or pharmacological manipulations to investigate which cortical areas are required for task-performance and decision-making [8,95]. In visual discrimination and decision-making tasks, flow of cortical activation is modulated by training in the task such that visual stimuli increases activity in higher visual areas which subsequently recruits activity in the secondary and primary motor cortices [127]. Incorporation of intersectional genetics during an odor discrimination task has additionally shown that specific subtypes of neurons contribute differentially to global task-related cortical activity [8]. Regional silencing using optogenetics or muscimol has shown that specific regions are critical in some tasks. For example, silencing of the anterolateral motor cortex abolishes licking behavior in an odor discrimination task [8], and visual cortex silencing impairs choice in a visual discrimination task [95]. The effects of regional silencing are task and region dependent with specific regions producing deficits in tasks with less cognitive demand, whereas tasks with high cognitive demand show deficits with inactivation across many cortical regions [95]. These studies highlight how mesoscopic Ca^2+^ imaging is providing new insights into higher cortical processing. 

### 6.3. Cortical Activity during Visual Processing

Wide-field mesoscopic imaging has facilitated advances in our understanding of visual processing in the mouse cerebral cortex. As for other behaviors, the processing of visual information engages large regions of the cerebral cortex. Imaging Ca^2+^ responses to moving visual stimuli reveal increased activity extending beyond primary visual cortex (V1) and established high visual areas, into postrhinal cortex (POR) and ectorhinal and temporal association cortices (ETC). However, the POR and ETC responses differ from V1 responses by lacking a retinotopic organization and reduced sensitivity to stimuli size, suggesting that POR and ETC are higher visual areas extending the putative ventral stream [128]. Although all visual areas respond to the coherence of visual stimuli movement, irrespective of the stimuli content, cortical regions in the putative dorsal stream have the highest activation, including the anterolateral (AL), posterior medial (PM), rostrolateral (RL), and mediolateral areas [129].

In a recent study, mesoscopic Ca^2+^ imaging of the dorsal cerebral cortex was used not only to monitor the cortical response to visual stimuli but to also provide real-time control in a closed feedback loop [125]. To reach a visual target using a cursor, mice were trained to de-correlate Ca^2+^ activity in two arbitrary cortical regions that would be highly correlated in naïve mice. In early training, V1 was activated during task execution compared to baseline (Figure 4D). During late training, higher visual areas including anteromedial cortex (AM), PM and RL are also recruited during task execution. Interestingly, when well-trained mice were presented with playback of the task execution, V1, AM, PM, and RL showed no activation (Figure 4D,E), arguing that subject agency differentiates how similar visual stimuli are processed.

### 6.4. Wide-Field Ca^2+^ Imaging in Neurological Disorders

New research avenues utilizing wide-field imaging are rapidly emerging, including investigations into neural dynamics in disease models as demonstrated in a mouse model of episodic ataxia type 2 with flavoprotein autofluorescence [130]. This includes advances using wide-field Ca^2+^ imaging and FC analyses. Recently, both local and global changes in mouse cortical networks were shown with Ca^2+^ imaging following an acute stroke induced by photothrombosis [131]. In the hemisphere contralateral to the stroke, FC increased between motor and sensory areas. In contrast, interhemispheric connectivity between synonymous cortical regions was reduced. Chronic post-stroke monitoring showed recovery of these changes in FC over time. A second study reproduced these changes in interhemispheric FC after a photothrombic stroke [132]. Intriguingly, micro-infarcts were insufficient to produce significant changes in cerebral cortical FC, while significantly impairing behavioral task performance.

Wide-field Ca^2+^ imaging is also being used to understand neuronal network dynamics in neurodegenerative disorders and epilepsy. Using the small molecule Ca^2+^ indicator OGB-1, long-range, slow wave activity was found to be de-correlated across the cerebral cortex in a murine model of Alzheimer’s disease [87,133]. Both cortical and subcortical regions exhibited hyperactivity, suggestive of a common excitatory/inhibitory imbalance [87,134]. The altered cortical connectivity, regional/cellular hyperactivity, as well as behavioral deficits, were rescued by systemic administration or direct cortical perfusion of benzodiazepines. In a mouse model of focal epilepsy, seizures propagated from the focus in V1 to both contiguous and homotopic regions prior to spreading to the rest of the cerebral cortex [135]. The seizure propagation appears to follow the established connectivity of the visual system, suggesting a hijacking of existing brain networks. These examples demonstrate the utility and promise of wide-field optical imaging to investigate a spectrum of preclinical disease/disorder models.

## 7. New Developments

### 7.1. Voltage Sensors

While Ca^2+^ imaging is a powerful approach to monitor neuronal activity, it does not measure changes in transmembrane potential, including action and synaptic potentials. As noted in the Introduction, voltage sensitive dyes suffered from low signal-to-noise and found limited use in in vivo preparations. Genetically encoded voltage indicators (GEVIs) should be ideal for this this type of neural imaging, with their direct voltage sensing properties and fast kinetics and a number of new and improved GEVIs emerging recently [136,137,138,139,140]. However, most of these GEVIs do not provide the sensitivity required to record neural activity in vivo during behaviors [141]. Use of an automated, high-throughput voltage screening platform changed the search strategy and identified GEVIs with improved sensor characteristics in the green (ArcLight and Marina) and red (FlicR2 and VARNAM) spectral wavelengths [142], including Ace-mNeon2, VARNAM2, and their respective reverse response polarity variants pAce and pAceR [143]. Using combinations of these GEVIs led to a novel, dual polarity multiplexing technique that exploits the multiphoton and orthogonal response polarities of the green and red indicator pairs (Ace-mNeon2 & pAce or VARNAM2 & pAceR). These 4 indicators have been targeted to distinct cell classes to record the concurrent voltage dynamics from as many as 4 neuron-types, as has been demonstrated in both cortical (V1) and sub-cortical (CA1) brain regions in mice using a custom dual-color microscope [143].

### 7.2. Combined Recording Techniques and Multimodal Sensing

Shortcomings of wide-field Ca^2+^ imaging sensors include a limited temporal resolution, and they are therefore, unable to capture the full range of frequency-specific information essential for many sensorimotor and cognitive behaviors. In addition, wide-field, single photon imaging, captures activity primarily from layers II and III of the cerebral cortex and the signal represents the combined neuropil activity in a region [71]. Combining simultaneous Ca^2+^ imaging with electrophysiological recordings from multiple brain regions would be a major advance, as would extending the spatial resolution to monitor single cell activity. A number of approaches have emerged to address these limitations and add to the functionality.

Several studies have combined two-photon (2P) imaging with wide-field imaging to provide single cell resolution [85,144]. As these techniques are still being developed, specialized equipment is required to perform wide-field 2P imaging. Wide-field 2P microscopy offers advantages over traditional wide-field imaging as thousands of neurons can be imaged simultaneously with cellular or subcellular resolution [145,146,147]. 2P imaging also allows increased depth of the imaging field and the potential for volumetric recordings. Newer wide-field 2P microscopes also offer rotation capabilities and long working distance, air immersion lenses that enable more imaging flexibility for different types of cranial windows, and the ability to image curved surfaces. Mesoscale 2P imaging is being used to confirm hypotheses tested in smaller neural populations, such as small-world structure and behavioral correlates of neural activity [146,148]. The high resolution and wide imaging field provided by mesoscale 2P imaging will provide significant insight into how microscale neural dynamics give rise to mesoscale and macroscale neural dynamics.

Wide-field imaging has also been combined with electrophysiological recordings, including in deeper structures [69,126]. These combined studies allow investigations into the relation between either local or distant single cell activity and mesoscopic Ca^2+^ dynamics. Simultaneous fMRI with wide-field Ca^2+^ imaging has also been accomplished, adding large scale cortical and subcortical BOLD activity [149]. Optogenetics have been combined with wide-field imaging to manipulate specific regions and circuits [95,126,150]. Systems have been developed for automated, self-initiated mesoscopic imaging in the animal’s home cage [151].

Advances have been made in transparent electrode arrays [152,153,154,155] that provide for simultaneous, multimodal recordings of Ca^2+^ fluorescence, and electrocorticography (ECoG). However, chronic studies with transparent ECoG electrodes have been restricted to a single brain region and a small fields of view (~2–5 mm^2^). To address this limited field of view, transparent, inkjet-printed electrode arrays have been integrated into morphologically conformant transparent polymer skulls (eSee-Shells) [77]. The electrodes and interconnects are composed of poly(3,4-ethylenedioxythiophene) polystyrene sulfonate (PEDOT:PSS), a transparent, printable conductor with excellent electrical properties. When implanted on GCaMP6f transgenic mice, eSee-Shells enable long duration, simultaneous mesoscale Ca^2+^ imaging and ECoG recordings over 45 mm^2^ of the dorsal cerebral cortex. When combined with sparse expression of GCaMP6s, single-cell Ca^2+^ fluorescence recordings are possible beneath the electrodes and interconnects while monitoring ECoG signals across the cortex.

### 7.3. Free Range Mesoscopic Ca^2+^ Imaging

Prior to the last decade, head-restraint of mice undergoing imaging was an experimental necessity. However, head-fixation is often problematic as animals can experience stress due to the condition, which alters neuronal physiology. Recent work looking at sensory processing in free and naturally behaving rodents, shows significant differences in neural activity as compared to similar experiments in head-fixed behaving mice. Additionally, many natural behaviors are impossible to perform during head-restraint such as social interaction, rearing, mating, foraging, and navigation. A key complementary development is the advent of machine learning-based marker-less tracking techniques for precisely tracking and quantifying complex behaviors seen in freely moving animals. Thus, technologies for cellular level recordings from microcircuits using miniaturized microscopes [156], and miniaturized tetrode devices [157] have become increasingly popular.

More recently, similar miniaturized devices have been developed that allow wide-field mesoscale imaging of much of the dorsal cortex of mice [158]. Using relatively simple imaging optics, these devices have integrated electronic hardware allowing multi-color excitation for fluorescence imaging and reflectance imaging for hemodynamic correction, and are designed to be docked quickly to transparent polymer skull implants [60,159]. The overall weight of these ‘mini-mScope’ devices is ~4 g and they can image an 88 mm^2^ field-of-view at resolutions ranging from 40 to 60 µm. These devices have enabled mesoscale imaging of cortical dynamics in mice freely exploring open fields, and socially interacting with companion mice. Further, this study demonstrated the ability to perform mesoscale imaging of glutamate dynamics in mice naturally transitioning from wakefulness to sleep. Such brain state changes, which are accompanied by large scale changes in brain activity, are hard to study in head-fixed conditions.

Future iterations of such mini-mScopes could potentially include faster image sensors [160] to perform voltage imaging across the cortex in mice expressing GEVIs specifically optimized for mesoscale single-photon imaging [140]. Such devices may reveal new insights into how mesoscale activity at temporal scales higher than those that can be captured using Ca^2+^ imaging mediate complex behaviors.

## 8. Conclusions

Recording neural dynamics at multiple spatial and temporal scales is required for a complete understanding of how the nervous system functions. The mesoscale level is particularly critical, as it is focused on the interactions occurring across different regions and bridges the expansive gap between micro and macroscale neural processing and network dynamics. Mesoscopic Ca^2+^ imaging has rapidly developed into one of the primary tools for monitoring the activity of neural networks in awake mice. As reviewed, mesoscopic Ca^2+^ imaging is being used to address a spectrum of questions about cerebral cortical processing in a variety of behaviors. Technological advancements in wide-field optical imaging are occurring at a rapid pace, including in genetically encoded Ca^2+^, and now voltage sensors, added functionalities such as optogenetics and multimodality recordings, targeting of specific neuronal populations, data analysis tools, and imaging in freely moving animals. These advances will ensure that mesoscopic optical imaging will continue to provide new insights and that the best is still to come.

## Figures and Tables

**Figure 1 biology-11-01601-f001:**
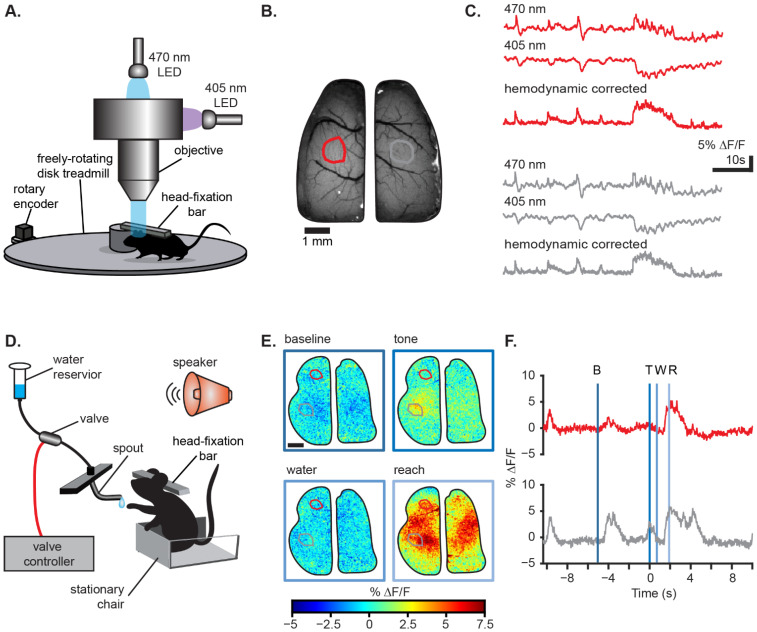
Wide-field Ca^2+^ imaging allows for investigation of neural processing during multiple behavioral paradigms. (**A**) Schematic showing an example of a wide-field imaging setup allowing for head-fixed behavior on a disk treadmill. (**B**) Example image showing the extent of the imaging field with non-brain regions masked in white. Scale bar 1 mm. (**C**) Example Ca^2+^ traces from ROIs (red and grey circles) in (**B**) during an imaging session on the disk treadmill setup in (**A**). Ca^2+^ traces shown are raw Ca^2+^-dependent GCaMP6f signals (470 nm; top), Ca^2+^-independent GCaMP6f signals (405 nm; middle), and hemodynamic corrected GCaMP6f signals (bottom). (**D**) Schematic showing an example of a wide-field imaging setup used for a cued reaching task. (**E**) Pseudo-colored images showing the change in hemodynamic corrected GCaMP6f fluorescence across the cortex during various stages of a cued reaching task. Scale bar: 1 mm. (**F**) Hemodynamic corrected Ca^2+^ traces for the ROIs (red and grey circles) shown in (**E**). Traces are centered with the auditory tone cue at time = 0. Blue lines denote the different stages of the reaching task matching the images in (**E**). B = baseline; T = tone; W = water delivery; R = reach.

**Figure 2 biology-11-01601-f002:**
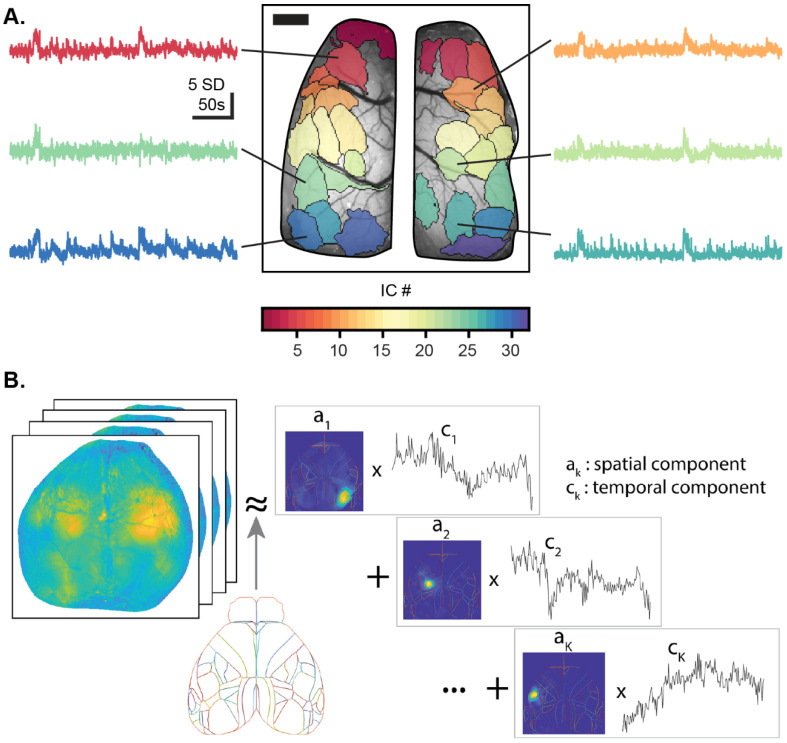
Overview of decomposition methods used to analyze wide-field Ca^2+^ imaging data. (**A**) Spatial independent components (ICs; colored regions) and time-courses (colored-traces) after independent component analysis of wide-field Ca^2+^ imaging data using the JADE algorithm superimposed on the cerebral cortical surface. Image scale bar = 1 mm. Time-courses z-scored with scale in standard deviations (SD). (**B**) Decomposition of a wide-field Ca^2+^ imaging video shown as a pseudo-color heat map into spatial components *a* (colored heat maps with atlas overlay), and temporal components *c*, with the spatial components soft-aligned to an atlas, here the Allen Institute Common Coordinate Framework (CCF) atlas. Panel (**B**) adapted with permission from [106].

**Figure 3 biology-11-01601-f003:**
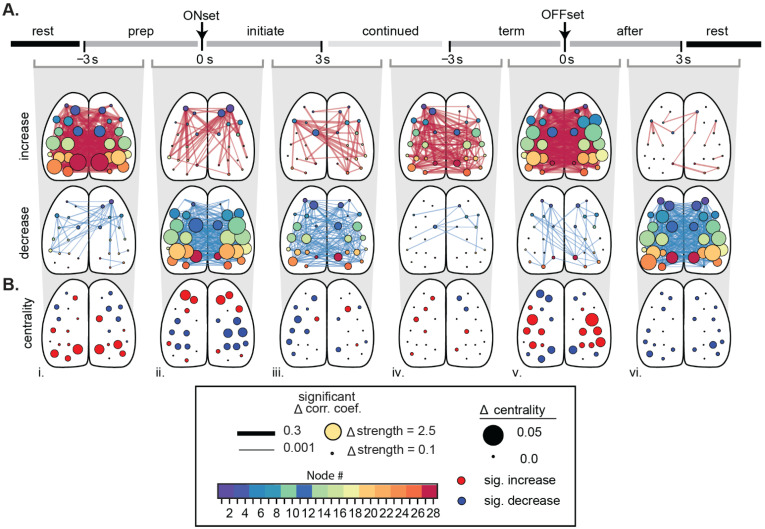
Cerebral cortical functional connectivity and centrality undergo large shifts from rest to locomotion and return to rest. (**A**) Significant changes in correlations between nodes during treadmill locomotion across sequential behavior periods (α < 0.05, with false discovery rate correction). The nodes and changes in functional connectivity are superimposed on the cortical surface. Significant increases shown in red (top) and decreases in blue (bottom). Directly above, the grey bracket indicates the adjacent behavior periods being compared (**i**–**vi**). The size of each node reflects the magnitude of significant increases or decreases, respectively. (**B**) Significant change in eigenvector centrality across the sequential behavior periods as in (**A**). Size of circles denote the magnitude of the change, while circle color is the direction of significant change (red = increase, blue = decrease, black = not significant; α < 0.05, with false discovery rate). Modified from [61] with permission.

**Figure 4 biology-11-01601-f004:**
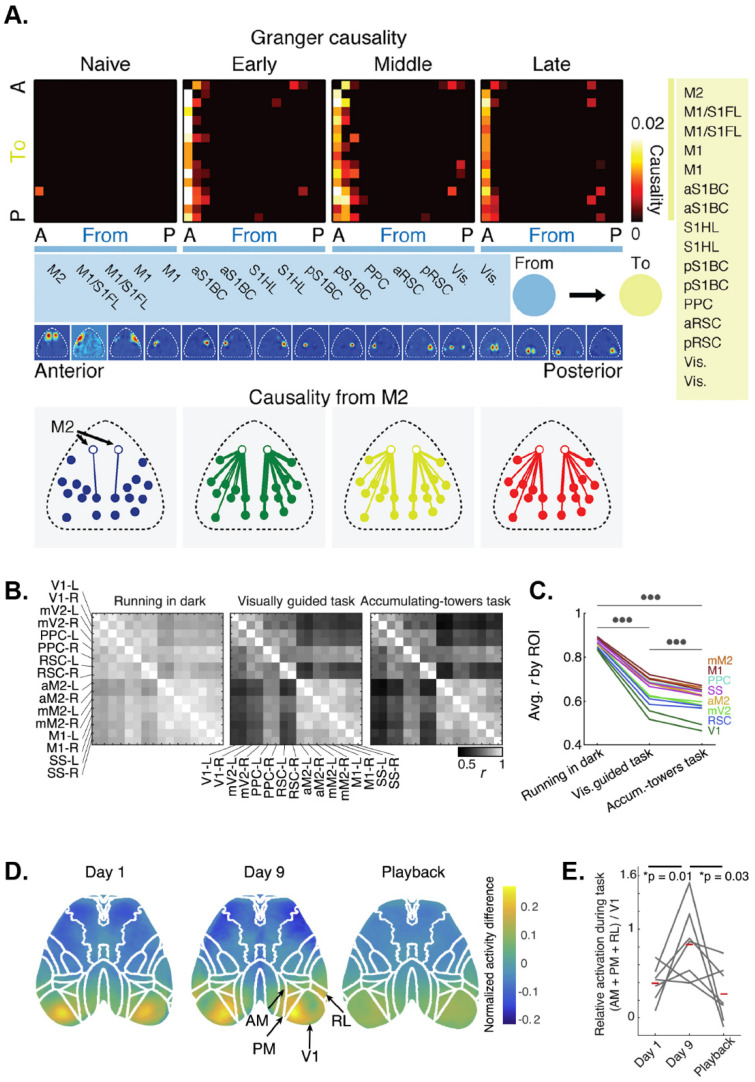
Multi-method investigation of cognitive processes using wide-field Ca^2+^ imaging. (**A**) Activity flow between M2 and other brain regions increases during motor learning. Matrices of granger causality values from M2 (left most column) to other brain regions (left to right, anterior to posterior) during four stages of learning of a motor task, in which mice are trained to press a lever beyond a specific threshold. Bottom row: spatial causality maps showing directionality from M2 to various regions are increased with learning. Modified with permission from [18]. (**B**) Correlation matrices showing differences in brain functional connectivity during different behavioral tasks requiring different levels of complexity. Panels (**B**,**C**) adapted with permission from [95]. (**C**) Line plot showing the average correlation coefficient per brain region between different behavioral paradigms (RM One-Way ANOVA *p* = 3.1 × 10^−23^ with post hoc Tukey’s multiple comparisons; *p* < 0.001; circles). (**D**) Cortical activation relative to baseline during a closed loop visual task. At Day 1 in task learning mice, cortical activation is limited to the primary visual cortex. In Day 9, task proficient mice, cortical activation extends to high visual areas. In task proficient mice, during passive task playback, there is no relative activation of the visual areas. (**E**) Activation of high visual areas relative to primary visual cortex increases in task proficient mice performing the task and is reduced during task playback. Panels (**D**,**E**) reused with permission from [125].

## Data Availability

Not applicable.

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
