# Peer review of "Wide-Field Calcium Imaging of Neuronal Network Dynamics In Vivo"

_biology, 2022, doi:10.3390/biology11111601_

Round 1

Reviewer 1 Report

The authors present a comprehensive and topical review of widefield calcium imaging techniques for the analysis of neuronal network dynamics.

The review is easily to follow, and I believe is an accurate snapshot of the field. Most of the major developments in widefield calcium imaging in mouse are covered and the chosen references are a good representation of the whole literature.

I only have a few minor comments:

General

1.     Most of the cited literature and key findings (widespread motor activations, locomotion, learning, etc.) come from studies done in mice. Nowadays there are several groups using widefield calcium imaging in other species, like macaques and marmosets. I believe this should be clarified early in the manuscript.

2.     The abstract mentions that the limitations of widefield imaging are examined; but these are only briefly mentioned on section 6.2 and in a few other places. The authors might consider summarizing the limitations in its own section: hemodynamics, temporal resolution, noise, light scattering, neurite contamination, “what are we really measuring”, etc.

Abstract

3.     I’m not sure what the authors mean by “disordered information processing”. I suppose it refers to the study of information flow in “health and disease”, but the term “disordered information processing” does not show up anywhere else in the manuscript.

1. Introduction

4.     Line 75: Refs [24-26] are quite specific. Consider citing a more general one. See for example Grienberger and Konnerth, Neuron 2012.

5.     Line 76: Regarding earliest Ca sensors I believe fura-2 and fluo-4 were much more impactful on the field than fluo-5f (that’s a little bit of nitpicking, sorry). They could also add some reference here (again, Grienberger and Konnerth could be a good one).

6.     Line 88: There’s been huge developments in the field in this area (cell-specific tagging for Ca imaging, viruses and transgenic lines). The authors might want to be a little more exhaustive in this section. See for example the works from the Allen (de Vries, Nat Neuro 2020), Couto et al (curr Ref 64), Kauvar et al (Neuron 2020) for sparse labeling, etc.

7.     Line 118: The authors might want to mention (here or somewhere else) that there are now widefield/mesoscale approaches using 2-photon imaging. See for example FASSHIO-2PM (Ota et al, Neuron 2021), and Diesel2p (Yu et al, Nat Com 2021). Even though these are single-cell resolution techniques, many of their findings correlate (or try to) with traditional widefield recordings. They will certainly be helpful for traditional widefield to correlate the micro- and meso- scale dynamics. There’s also mesoscale studies trying to correlate between meso and synaptic resolution (Lu et al, Nature Methods 2020).

6.2 Combined recording techniques and multimodal sensing

8.     Line 446: I’m not sure Ref 120 has widefield imaging (other than for the retinotopies). Maybe Zatka-Hass et al, eLife 2021 from the same group? Or Peters et al (ref 72) ?

Author Response

Reviewer 1 Comments

 We would like to thank the reviewers for both their positive comments and constructive suggestions. We have made every effort to address their concerns, as detailed below (italicized font). All changes are highlight in underlined, red text in the manuscript file.

 The authors present a comprehensive and topical review of widefield calcium imaging techniques for the analysis of neuronal network dynamics. The review is easily to follow, and I believe is an accurate snapshot of the field. Most of the major developments in widefield calcium imaging in mouse are covered and the chosen references are a good representation of the whole literature.

We appreciate the positive comments by the reviewer and thank them for their feedback.

General

  1. Most of the cited literature and key findings (widespread motor activations, locomotion, learning, etc.) come from studies done in mice. Nowadays there are several groups using widefield calcium imaging in other species, like macaques and marmosets. I believe this should be clarified early in the manuscript.

As suggested, we have added that wide-field imaging is being performed in other species, including macaques and marmosets, at line 79.

 “More recently, wide-field Ca2+ imaging is being explored in different species, including rats [26,27], marmosets [28-30], and macaques [31,32], with an emphasis on single cell resolution.” 

  1. The abstract mentions that the limitations of widefield imaging are examined; but these are only briefly mentioned on section 6.2 and in a few other places. The authors might consider summarizing the limitations in its own section: hemodynamics, temporal resolution, noise, light scattering, neurite contamination, “what are we really measuring”, etc.

We have added a new section, “Limitations of wide-field imaging” at line 174.  We have tried to address all the issues that this reviewer and Reviewer 2 have raised.

 “4. Limitations of wide-field imaging

As with any technique, mesoscopic imaging has limitations. Based on single photon (1P) epifluorescence imaging, the mesoscale approach is restricted to recording activity near and below the surface of brain. For example, in the cerebral cortex, the Ca2+ fluorescence signals are primarily from neuronal activity in layers II/III [65,71]. The depth of recording for 1P imaging in dense tissues can be improved, to some degree, with red-shifted sensors [50,72], as longer wavelengths penetrate tissue better and are prone to less scatter. In some small systems with unimpeded optical access, such as larval zebrafish, imaging the entire brain using light sheet microscopy can allow for volumetric functional wide-field single-cell Ca2+ imaging [73,74]. This is extremely challenging to perform in systems other than zebrafish or fixed tissue.

Due to light scatter produced by imaging through dense tissue and the low numerical aperture of mesoscopic lenses, mesoscopic imaging cannot provide high lateral or axial resolution as with other forms of microscopy [75]. Instead, the observed neural signal is a population summation of the activity within essentially a 3-dimensional voxel (not just 2 dimensional) that includes the dendrites, somata, and axons within that volume. While typical wide-field Ca2+ imaging does not report the activity of single cells, there are strategies, such as sparse expression of indicators and/or restriction to the soma, that can provide singe cell resolution [76,77].

In addition, the temporal resolution of mesoscale imaging is limited by the kinetics of the GCaMP protein. While the sensitivity and speed of GECIs have been vastly improved in recent years with targeted mutations, their decay time for single action potentials is still greater than 100 ms and their ability resolve high frequency firing with concurrent increases in fluorescence amplitude is also limited by both saturation and kinetics [38,50,75]. The timescale of GCaMPs are considerably slower than those of action potentials, which are on the order of 2-5 ms [78]. This difference in timescale between action potentials and Ca2+ fluorescence, especially in cases where single cells can be isolated, has led to the development of algorithms to reconstruct underlying spike trains from recorded Ca2+ signals [79-81] and methods of simultaneous electrophysiological and Ca2+ recordings [38,40]. The accuracy of inferred spike trains from Ca2+ data and construction of neural networks is dependent upon the method of reconstruction used and the experimental noise level [82,83]. These confounds can influence network connectivity and spike inference results and need to be considered during interpretation of results.

As discussed above, the primary family of Ca2+ indicator used, GCaMP, can be contaminated by both hemodynamics [8,65,67] and other Ca2+-independent fluorescence changes such as flavoprotein autofluorescence [58,66] requiring a correction strategy such as dual wavelength imaging. Additional contamination can arise from brain motion, baseline signal fluctuations, rhythmic physiological processes, and photonic noise (i.e. variability in the emission/detection of photons) [79]. Noise sources are a significant confounding factor in spike estimation, but newer algorithms are being improved to accommodate additional noise sources [84]. While the limitations of wide-field Ca2+ are not to be underestimated, its utility in revealing signaling relation-ships between brain areas remains invaluable.”

Abstract

  1. I’m not sure what the authors mean by “disordered information processing”. I suppose it refers to the study of information flow in “health and disease”, but the term “disordered information processing” does not show up anywhere else in the manuscript.

We have changed the wording of this sentence in the Abstract to better describe the discussion of neural processing in disease at line 25.

 “Also discussed is how wide-field Ca2+ imaging is providing novel insights into both normal and altered neural processing in disease.”

  1. Introduction
  2. Line 75: Refs [24-26] are quite specific. Consider citing a more general one. See for example Grienberger and Konnerth, Neuron 2012.

We have added the requested citation at line 85.

  “Ca2+ concentration provides a fast readout of changing neural activity, although indirect, as it is involved with numerous intracellular signaling cascades, intracellular Ca2+ storage and release, and synaptic transmission [33-36].”

  1. Line 76: Regarding earliest Ca sensors I believe fura-2 and fluo-4 were much more impactful on the field than fluo-5f (that’s a little bit of nitpicking, sorry). They could also add some reference here (again, Grienberger and Konnerth could be a good one).

We have modified the sentence at line 88 accordingly.

 “Some of the earliest Ca2+ sensors were dyes, such as BAPTA, fura-2, and fluo-4 [36].”

  1. Line 88: There’s been huge developments in the field in this area (cell-specific tagging for Ca imaging, viruses and transgenic lines). The authors might want to be a little more exhaustive in this section. See for example the works from the Allen (de Vries, Nat Neuro 2020), Couto et al (curr Ref 64), Kauvar et al (Neuron 2020) for sparse labeling, etc.

We have expanded the referenced statement at line 99 to include the requested studies.

 “Additionally, dual expression systems in combination with intersectional genetics and/or promoter specific viruses can be used to express sensors in specific populations or subpopulations of cells [43-45]. Utilization of an ever-expanding number of Cre-driver lines has revealed a litany of subtle response properties in visual cortical neurons [46]. Conversely, strategies for driving sparse expression of Ca2+ indicators allow for single cell resolution at the mesoscale level [47,48].”

 Line 118: The authors might want to mention (here or somewhere else) that there are now widefield/mesoscale approaches using 2-photon imaging. See for example FASSHIO-2PM (Ota et al, Neuron 2021), and Diesel2p (Yu et al, Nat Com 2021). Even though these are single-cell resolution techniques, many of their findings correlate (or try to) with traditional widefield recordings. They will certainly be helpful for traditional widefield to correlate the micro- and meso- scale dynamics. There’s also mesoscale studies trying to correlate between meso and synaptic resolution (Lu et al, Nature Methods 2020).

As suggested, we have additional information to section 7.2 “Combined recording techniques and multimodal sensing” at line 504.

 “Several studies have combined two-photon (2P) imaging with wide-field imaging to provide single cell resolution [85,144]. As these techniques are still being developed, specialized equipment is required to perform wide-field 2P imaging. Wide-field 2P microscopy offers advantages over traditional wide-field imaging as thousands of neurons can be imaged simultaneously with cellular or subcellular resolution [145-147]. 2P imaging also allows increased depth of the imaging field and the potential for volumetric recordings. Newer wide-field 2P microscopes also offer rotation capabilities and long working distance, air immersion lenses that enable more imaging flexibility for different types of cranial windows, and the ability to image curved surfaces. Mesoscale 2P imaging is being  used to confirm hypotheses tested in smaller neural populations, such as small-world structure and behavioral correlates of neural activity [146,148]. The high resolution and wide imaging field provided by mesoscale 2P imaging will provide significant insight into how microscale neural dynamics give rise to mesoscale and macroscale neural dynamics.”

 6.2 Combined recording techniques and multimodal sensing

  1. Line 446: I’m not sure Ref 120 has widefield imaging (other than for the retinotopies). Maybe Zatka-Hass et al, eLife 2021 from the same group? Or Peters et al (ref 72).

The reviewer is correct as Reference 120 does not involve mesoscopic imaging. We have replaced with the suggested Zatka-Hass eLife 2021 reference at line 518.

Reviewer 2 Report

This is a very interesting review that summarizes and discusses the state of the art in wide-field microscopy. The article is clear, well organized and written. I have suggestions for improvement that I think are important to provide a complete picture and make the review even more attractive.

1. The title may lead to confusion, since it applies to both in vivo and in vitro systems. Maybe “Wide-field Calcium Imaging of Neuronal Network Dynamics in vivo”?

2. As a review, I think it’s important to briefly mention that WF microscopy has been extensively used in the study of neuronal cultures, which is by itself a gigantic focus of research, and that research in cultures often completes the one in vivo, e.g., to understand the emergence of universal mechanisms or test several issues associated with the processing of calcium data. Maybe the authors, at the end of the Introduction, can add something like: “Calcium imaging has been also used in in vitro systems, particularly neuronal cultures, to study the development and functional organization of de novo formed neuronal networks. Although research in neuronal cultures is extensive, we exclude them here since they shape neuronal systems that have lost the native architecture of the brain and are typically isolated from inputs”, and maybe comment on the use in vitro systems to address Ca2+ problems (see my next point).

3. As a new section, or as an extension of sections 2, 3 o 4, the authors should describe one of the major limitations of Ca2+ imaging, namely that the dynamics of the fluorescence probe masks the details of the spike trains, so one needs to develop quite complex numerical algorithms to infer (and often just roughly predict) the underlying spike trains of a calcium signal. This true nightmare has motivated many people to measure simultaneously Ca2+ signals and spike trains (see P. Jercog et al., Large-Scale Fluorescence Calcium-Imaging Methods for Studies of Long-Term Memory in Behaving Mammals, 2016) or to develop algorithms to infer spikes trains (Oasis, Peeling, ML-Spike,… see e.g. T. Deneux et al., Nature Communications 2016; Grewe et al., Nat. Methods 2010; and others). The problem is very important, since the results of an investigation may depend on the algorithm used to extract the spikes (Tibau et al., IEEE TRANSACTIONS ON NETWORK SCIENCE AND ENGINEERING, 2020; Göbel and Helmchen, Physiology 2007). The problem is also important in the context of functional connectivity. For instance, one can compute functional connectivity from either the fluorescence traces or from the inferred spikes. The comparison of both can be helpful to verify that results are consistent.

4. For completeness, I think that the authors should explain, either in the Introduction or Section 6, the advantages and drawbacks of wide-field microscopy as compared to more advanced techniques such as light-sheet microscopy (e.g. Bernardello et al., “Modular multimodal platform for classical and high throughput light sheet microscopy”, Sci Rep 2022, and others). The application of LSM to Ca2+ brain zebrafish imaging, retina and other systems has caught substantial interest. For instance, one may argue that wide-field is relatively cheap and can access large fields of view, while LSM is very expensive and the accessible volume, although highly valuable, is still small.

Author Response

Reviewer 2 Comments

We would like to thank the reviewers for both their positive comments and constructive suggestions. We have made every effort to address their concerns, as detailed below (italicized font). All changes are highlight in underlined, red text in the manuscript file.

This is a very interesting review that summarizes and discusses the state of the art in wide-field microscopy. The article is clear, well organized and written. I have suggestions for improvement that I think are important to provide a complete picture and make the review even more attractive.

We thank the reviewer for their positive comments.

  1. The title may lead to confusion, since it applies to both in vivo and in vitro systems. Maybe “Wide-field Calcium Imaging of Neuronal Network Dynamics in vivo”?

As suggested, we have altered the title to specify that the review applies to in vivo systems: “Wide-field Calcium Imaging of Neural Network Dynamics In Vivo” at line 2-3.

  1. As a review, I think it’s important to briefly mention that WF microscopy has been extensively used in the study of neuronal cultures, which is by itself a gigantic focus of research, and that research in cultures often completes the one in vivo, e.g., to understand the emergence of universal mechanisms or test several issues associated with the processing of calcium data. Maybe the authors, at the end of the Introduction, can add something like: “Calcium imaging has been also used in in vitro systems, particularly neuronal cultures, to study the development and functional organization of de novo formed neuronal networks. Although research in neuronal cultures is extensive, we exclude them here since they shape neuronal systems that have lost the native architecture of the brain and are typically isolated from inputs”, and maybe comment on the use in vitro systems to address Ca2+ problems (see my next point).

As suggested we have added a statement noting work in in vitro systems at line 76.

“Ca2+ imaging has been extensively used in in vitro systems, particularly neuronal cultures, to study the development and functional organization of de novo network formation and test the properties of Ca2+ sensors [24,25], and in vivo to study intrinsic network dynamics.”

  1. As a new section, or as an extension of sections 2, 3 o 4, the authors should describe one of the major limitations of Ca2+ imaging, namely that the dynamics of the fluorescence probe masks the details of the spike trains, so one needs to develop quite complex numerical algorithms to infer (and often just roughly predict) the underlying spike trains of a calcium signal. This true nightmare has motivated many people to measure simultaneously Ca2+ signals and spike trains (see P. Jercog et al., Large-Scale Fluorescence Calcium-Imaging Methods for Studies of Long-Term Memory in Behaving Mammals, 2016) or to develop algorithms to infer spikes trains (Oasis, Peeling, ML-Spike, see e.g. T. Deneux et al., Nature Communications 2016; Grewe et al., Nat. Methods 2010; and others). The problem is very important, since the results of an investigation may depend on the algorithm used to extract the spikes (Tibau et al., IEEE TRANSACTIONS ON NETWORK SCIENCE AND ENGINEERING, 2020; Göbel and Helmchen, Physiology 2007). The problem is also important in the context of functional connectivity. For instance, one can compute functional connectivity from either the fluorescence traces or from the inferred spikes. The comparison of both can be helpful to verify that results are consistent.

As discussed above in our response to Reviewer 1, we have added a new section at line 174 on the Limitations of Wide-field Imaging, that addresses these comments as well as those raised by Reviewer 1.

  1. For completeness, I think that the authors should explain, either in the Introduction or Section 6, the advantages and drawbacks of wide-field microscopy as compared to more advanced techniques such as light-sheet microscopy (e.g. Bernardello et al., “Modular multimodal platform for classical and high throughput light sheet microscopy”, Sci Rep 2022, and others). The application of LSM to Ca2+ brain zebrafish imaging, retina and other systems has caught substantial interest. For instance, one may argue that wide-field is relatively cheap and can access large fields of view, while LSM is very expensive and the accessible volume, although highly valuable, is still small.

As suggested by both this reviewer and Reviewer 1, we have added a new section (section 4) on the limitations of wide-field imaging which addresses this issue with specific reference to LSM at line 181.
